# Vector control for malaria prevention during humanitarian emergencies: protocol for a systematic review and meta-analysis

Louisa Alexandra Messenger  , Joanna Furnival-Adams, Bethanie Pelloquin, Mark Rowland

Disease Control, London School of Hygiene & Tropical Medicine, London, UK

**Correspondence to**
Dr Louisa Alexandra Messenger; louisa.messenger@lshtm.ac.uk

## ABSTRACT

**Introduction** Humanitarian emergencies, of either natural or anthropogenic origins, are equivalent to major disasters, which can lead to population displacement, food insecurity and health system disruptions. Almost two-thirds of people affected by humanitarian emergencies inhabit malaria endemic regions, particularly the WHO African Region, which currently accounts for 93% and 94% of malaria cases and deaths, respectively. As of late 2020, the United Nations Refugee Agency estimates that there are globally 79.5 million forcibly displaced people, including 45.7 million internally displaced people, 26 million refugees, 4.2 million asylum-seekers and 3.6 million Venezuelans displaced abroad.

**Methods and analyses** A systematic review and meta-analysis will be conducted to evaluate the impact of different vector control interventions on malaria disease burden during humanitarian emergencies. Published and grey literatures will be systematically retrieved from 10 electronic databases and 3 clinical trials registries. A systematic approach to screening, reviewing and data extraction will be applied based on the Preferred Reporting Items for Systematic Reviews and Meta-Analysis guidelines. Two review authors will independently assess full-text copies of potentially relevant articles based on inclusion criteria. Included studies will be assessed for risk of bias according to Cochrane and certainty of evidence using the Grading of Recommendations Assessment, Development and Evaluation (GRADE) approach. Eligible studies with reported or measurable risk ratios or ORs with 95% CIs will be included in a meta-analysis. Subgroup analyses, including per study design, emergency phase and primary mode of intervention, may be performed if substantial heterogeneity is encountered.

**Ethics and dissemination** Ethical approval is not required by the London School of Hygiene and Tropical Medicine to perform secondary analyses of existing anonymous data. Study findings will be disseminated via open-access publications in peer-reviewed journals, presentations to stakeholders and international policy makers, and will contribute to the latest WHO guidelines for malaria control during humanitarian emergencies.

**PROSPERO registration number** CRD42020214961.

### Strengths and limitations of this study

► This will be the first systematic review and meta-analysis of the impact of different vector control interventions on malaria disease burden during humanitarian emergencies, providing a comprehensive global evidence summary that can be updated as new evidence emerges.

► Inclusion of a broad range of study types will allow the identification of studies from a diverse range of humanitarian contexts.

► By definition, humanitarian emergencies are unpredictable and volatile and consequently, these settings can be restrictive with respect to study design, limiting the number of community-level randomised control trials with epidemiological outcomes which can be considered in this review.

► Lack of a sensitive literature search strategy may result in a large number of titles and abstracts to screen initially.

## INTRODUCTION

### Malaria in humanitarian emergencies

Malaria is a life-threatening parasitic disease caused by *Plasmodium* species and transmitted by female *Anopheles* mosquitoes.[1] In 2018, there were an estimated 228 million cases of malaria worldwide, including 405 000 deaths.[1] As defined by the WHO, a humanitarian emergency is a calamitous situation in which the functioning of a community or society is severely disrupted, causing human suffering and material loss that exceeds the affected population's ability to cope using its own resources.[2] Both natural (eg, earthquakes, floods or tsunamis) and anthropogenic (eg, violent political conflicts) hazards can lead to humanitarian emergencies, which may involve population displacement, food insecurity and disruptions to health systems; all of these factors contribute to excess mortality and morbidity among affected communities. While natural disasters may be short-lived,

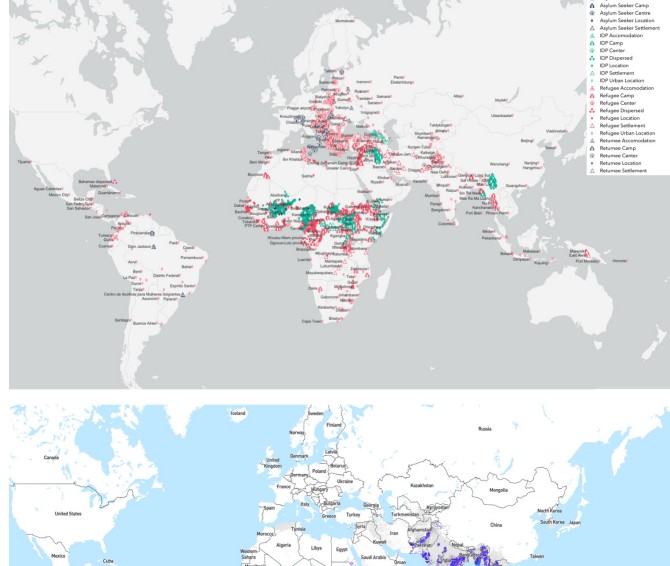

**Figure 1** Top: global map of internally displaced people (green) and refugee (red) populations, 2020. The UNHCR currently estimates that there are 79.5 million forcibly displaced people, including 45.7 million internally displaced people, 26 million refugees, 4.2 million asylum-seekers and 3.6 million Venezuelans displaced abroad.[3] Bottom: global map of *Plasmodium falciparum* parasite rate in children aged 2–10 years, 2000–2017.[54]

complex emergencies are often protracted, characterised by having complex social, political and economic origins, breakdown of governmental authority and infrastructures and often armed political conflict and human rights abuses.[2] Globally, the United Nations Refugee Agency

(UNHCR) currently estimates there are 79.5 million forcibly displaced people[3]; with almost two-thirds of people affected by humanitarian emergencies inhabiting malaria endemic regions[4] (figure 1).

Mass population displacement can increase the risk of epidemics and incidence of severe malaria, especially when immunologically naive individuals with little to no prior malaria exposure move to areas of more intense transmission. These populations are rendered vulnerable to epidemic malaria by their lack of protective immunity, increased concentration of people in exposed settings, inadequate health services, breakdown of national malaria control programmes, loss or emigration of skilled health-care workers, insufficient access to effective treatment and control measures, food scarcity, and concomitant infections, malnutrition and anaemia.[2] During humanitarian emergencies, malaria morbidity and mortality can contribute to the breakdown of health services and local case management, which can both worsen and be a source of the emergency.

## Humanitarian emergency features and settings

Humanitarian emergencies can be classified by their speed of onset, duration and setting, which all have implications for infectious disease control strategies (figure 2). Rapid-onset and slow-onset emergencies describe a sudden or gradual deteriorating situation, respectively, while protracted (chronic) emergencies involve consistently high levels of humanitarian need. The criteria of the acute phase (phases 1 and 2 in figure 2) are elevated mortality rates, poor access to effective healthcare for the affected population, response needs that are beyond local or national capacity, and possible breakdown of normal coordination and response mechanisms. The post-acute phase (phase 3) includes a reduction in peak mortality rates and improved access to humanitarian aid and healthcare. The recovery phase (phase 4) is characterised by the resumption of government, the withdrawal of humanitarian agencies and introduction of United

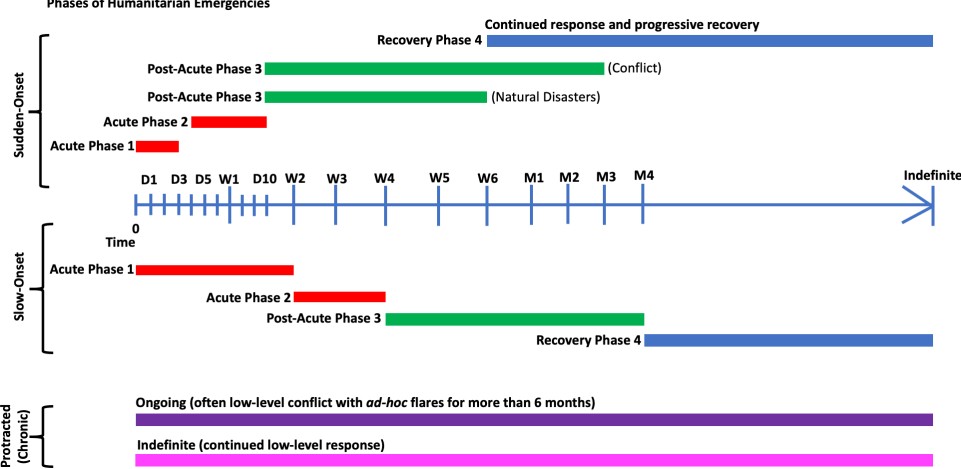

**Figure 2** Phases of sudden-onset, slow-onset or protracted (chronic) humanitarian emergencies. Time is indicated in days (D), weeks (W) and months (M). Adapted from World Health Organization.[2]

**Table 1** Key factors influencing choice of vector control intervention(s) during humanitarian emergencies

| Key factors | Subfactors |
|---|---|
| Malaria epidemiology | Level of endemicity; local transmission dynamics and stability; circulating *Plasmodium* species and proportions of coinfection; level of drug resistance; asymptomatic infections |
| Human population composition and behaviour | Population origin and size; displacement dynamics and mobility; level of prior malaria exposure; proportion of vulnerable individuals (eg, pregnant women, children and the elderly); age and sex distributions; sleeping location and night-time behaviour; knowledge of malaria; education; healthcare and treatment seeking behaviour; prevalence of comorbidities (eg, malnutrition, anaemia, HIV); gender mobility and work practices |
| Vector species distribution and behaviour | Local vector species composition; vector seasonality; host preference (anthropophillic/zoophillic); feeding time and location (early/late biting and indoor/outdoor biting); resting location (indoor/outdoor resting); level of insecticide resistance; flight range; breeding site preference |
| Emergency setting and infrastructure | Type of shelter or housing available (eg, ad-hoc refuse materials, plastic sheeting, tents, more permanent housing); road access and condition; camp or community organisation and layout; prior and present vector control interventions or programmes; timing of rainy seasons; potential to impact other vector-borne diseases (such as in areas of sympatric leishmaniasis) and nuisance insects (eg, lice, fleas, ticks, biting flies, bed bugs) |

Nations and non-governmental organisation (NGO) programmes and agencies. However, returning refugees may also re-integrate, bringing malaria with them, and depending on the level of strategic interest, this phase may or may not be neglected internationally. Protracted emergencies can remain entrenched in the post-acute/early-recovery phases or cycle between acute and post-acute phases indefinitely.

## Malaria vector control during humanitarian emergencies

During the acute phase of a humanitarian emergency, the first priorities for malaria control are effective case management, with prompt diagnosis and treatment.[2] Choice of vector control intervention will be determined by a number of key factors including, malaria infection risk and transmission dynamics, population demographics, prior malaria exposure, behaviour and mobility, vector species composition, vector feeding and resting preferences and emergency settings and infrastructure (table 1).

WHO recommended malaria control interventions, insecticide-treated nets (ITNs) and indoor residual spraying (IRS), are usually the first line preventative measures considered. Effective case management can be supplemented with long-lasting insecticidal net (LLIN) distributions, first targeting the most vulnerable populations (eg, pregnant women and children under-5 years old), with the goal of achieving and maintaining universal community-level coverage.[5–19] IRS can be applied in well-organised, politically stable settings, such as transit camps, but is generally not feasible when dwellings are scattered widely, of a temporary nature, or constructed with surfaces that are unsuitable for spraying.[20] IRS may be more appropriate for protecting larger refugee populations in more compact and stable settings where housing is more permanent and structurally sound, and there is an absence of alternative resting sites.[21–23] Targeted IRS can also supplement LLINs and be used together for insecticide resistance management.[24 25]

A suite of other community-level and personal vector control interventions has been designed for deployment during humanitarian emergencies, which will be considered in this review. One strategy has been to exploit utilitarian shelter materials, often provided in the early stages of emergencies, such as plastic sheeting, tarpaulins and tents, as mechanisms of insecticide delivery.[26–29] Additional opportunities identified to apply insecticide among at-risk populations have included distributing insecticide treated bed sheets and blankets, clothing, hammocks and wall lining,[30–38] and use of topical insecticides for human populations[39–41] or to treat community livestock.[42]

Most interventions act by delivering a lethal dose of insecticide to the adult mosquito, protecting the host from exposure to a potentially infective bite. Some of these interventions, notably LLINs and IRS, may also have a community wide benefit (in areas of vector susceptibility to the active ingredients), decreasing mosquito density and longevity, and ultimately interrupting malaria transmission. The insecticide used is dependent on the intervention and resistance profiles of local vector species.[43 44] Other interventions, such as topical repellents, work by preventing mosquitoes from detecting the host, by disrupting olfactory reception.[45] These interventions only provide individual protection and do not impact vectorial capacity.[43]

There are multiple factors involved in ensuring a successful vector control programme during humanitarian emergencies. The intervention(s) selected must be feasible and consider local vector species, human resources, funding, logistics and population mobility.[2] Other benefits of vector control interventions, which may make them more acceptable to the user, include reducing flies in households and nuisance biting.[46] and improved health and yields of treated livestock.[42] Some interventions may have low acceptability, for example, sleeping under an ITN or treated blanket/bedsheet in hot, humid conditions, some types of treated clothing may

be culturally inappropriate, topical repellents may cause irritation to individuals, and there may be a decrease in adherence over time to interventions which require sustained behavioural changes.[40]

### Why it is important to do this review

To date, there has been insufficient, systematic syntheses of evidence for the WHO to make recommendations for deployment of malaria vector control interventions in humanitarian settings. While a number of aforementioned interventions have either been evaluated during or developed specifically for use in humanitarian emergencies, the often unpredictable and volatile nature of these settings can often be restrictive in terms of designing experimental studies. Given these logistical constraints, a range of both observational and experimental study designs will be considered.

### Objectives

This study aims to evaluate the impact of different vector control interventions on malaria disease burden during humanitarian emergencies.

Primary objective is
▶ To evaluate the impact of different vector control interventions on malaria disease burden during humanitarian emergencies.

Secondary objectives are
▶ To explore whether the impact of different vector control interventions varies with the phase of humanitarian emergency (acute/post-acute, protracted and recovery/post-conflict phases).
▶ To explore whether the impact of different vector control interventions varies with the level of malaria transmission (low, <50% parasite rate in children; and high, >50% parasite rate in children).
▶ To explore whether the impact of different vector control interventions varies with the level of coverage during a humanitarian emergency.

### METHODS AND ANALYSES

The following types of studies will be considered for inclusion in this review:
▶ Cluster-randomised controlled trials (c-RCTs).
▶ Cluster-randomised cross-over studies.
▶ Cluster-randomised studies using a stepped-wedge design.
▶ Controlled before-and-after studies.
▶ Cohort studies (prospective or retrospective).
▶ Non-randomised cross-over studies.
▶ Programmatic evaluations.
▶ Cross-sectional studies.
▶ Case–control studies.
▶ Case series.
▶ Interrupted time series.

Experimental hut trials, other phase II studies or user acceptability surveys (including, quantitative knowledge, attitude and practices questionnaires or qualitative

observations, in-depth interviews and focus group discussions) will be summarised descriptively.

### Type of participants

Individuals of all age groups affected by humanitarian emergencies in malaria endemic regions; including those that have been internally displaced by humanitarian emergencies as well as host communities. A humanitarian emergency is defined as a calamitous situation, of either natural (including famine, flooding (localised or widespread), storms, cyclones, earthquakes, tsunamis) or anthropogenic origin (may include widespread (ie, affecting whole regions/countries) or geographically confined conflict (ie, affecting only parts of a country/or multiple countries, eg, the West African Islamic terrorist crisis affecting border regions of Mali, Niger, Nigeria, Burkina Faso, Cameroon and Chad), or persecution of a specific ethic group (eg, Bosnian, Rohingya and Rwanda genocides), in which the functioning of a community or society (whether a subsection or whole population) was severely disrupted, causing human suffering and material loss that exceeded the affected population's ability to cope using its own resources.[2] Studies with military personnel or peacekeepers, in demilitarised zones, with migrant, tribal or nomadic populations or temporary/seasonal labourers (eg, forest or dam workers, gold or gem miners) will be excluded.

### Type of interventions

Any malaria-specific vector control intervention, which has been evaluated during a humanitarian emergency, will be considered for inclusion (table 2).

### Type of outcome measures

To be eligible for inclusion, studies must report at least one of the following primary outcomes.

### Primary outcomes
▶ Malaria case incidence: measured as number of cases per unit of time or the number of new uncomplicated malaria cases. We will use site-specific definitions as long as they have demonstrated (1) a fever or history of fever and (2) confirmed parasitaemia (by blood smear microscopy, rapid diagnostic test (RDT) or PCR).
▶ Malaria infection incidence: measured as count per person per unit of time or the number of new infections (individuals must have confirmed parasitaemia by blood smear, RDT or PCR).
▶ Parasite prevalence (clinical and subclinical malaria): the proportion of surveyed individuals with confirmed parasitaemia at the community-level.

### Secondary outcomes
#### Epidemiological
▶ All-case mortality.
▶ Severe malaria, based on confirmed malaria diagnosis (individuals must have confirmed parasitaemia by blood smear, RDT or PCR) plus the presence of

**Table 2** Types of intervention to be included in this review

| Intervention category | Examples of intervention | Control* |
|---|---|---|
| Insecticide-treated temporary housing materials | Plastic sheeting; tarpaulins; tents | Untreated temporary housing material |
| Insecticide-treated bedding materials | Sheets; blankets; chaddars; top sheets | Untreated bedding material |
| Insecticide-treated wall linings | Plastic woven or non-woven mesh material used to cover housing structure/shelters | Untreated wall lining |
| Insecticide-treated clothing | Military uniforms; occupational clothing | Untreated clothing |
| Insecticide-treated curtains | Netted curtain or window material | Untreated curtains |
| Insecticide treatment of livestock | Cattle dipping; use of endectocides | No cattle dipping/ endectocide use |
| Personal repellents | Spatial repellents; topical repellents; insecticidal soaps | No repellent |
| Bed nets | Long-lasting insecticidal nets; insecticide-treated nets; untreated nets | Untreated bed nets (when compared with insecticidal counterparts) |
| Indoor residual spraying | Chemical treatment of housing structure/ shelters | No chemical treatment of housing structures/ shelters |
| Larval source management | Chemical or biological treatment of breeding sites; habitat modification or manipulation | No insecticidal or mechanical control |

*Studies where the control group comprises no other insecticidal malaria vector control intervention will also be considered. Use of other vector control tools (in addition to the intervention under evaluation) must be balanced between study arms.

clinical or laboratory features of vital organ dysfunction (these include impaired consciousness, coma, convulsions, respiratory distress, shock (systolic blood pressure <70 mm Hg in adults, <50 mm Hg in children), jaundice, haemoglobinuria, hypoglycaemia, severe metabolic acidosis and anaemia).

▶ Anaemia prevalence as per WHO cut-offs, based on haemoglobin measurements taken in community surveys.

### Entomological
▶ Entomological inoculation rate: the estimated number of bites by infectious mosquitoes per person per unit of time. This is calculated using the human biting rate (the number of mosquitoes biting an individual over a stated period, measured directly using human baits or indirectly using light traps, knock-down catches, entry or exit traps, baited huts or other methods of biting rate determination) multiplied by the sporozoite rate.

▶ Adult mosquito density: measured by a technique previously shown to be appropriate for the particular vector species (eg, human baits, light-traps, knock-down catches, entry or exit traps, baited huts or other methods).

▶ Sporozoite rate: measured as the number of caught adult mosquitoes positive for malaria sporozoites. Sporozoites can be detected through either molecular or immunological methods.

### Operational
▶ Intervention durability: measured by a technique previously shown to be appropriate, (eg, a time-series of WHO cone bioassays to monitor IRS residual activity or ITN/LLIN bioefficacy plus High Performance Liquid Chromatography (HPLC) to determine intervention insecticidal content).

### Adverse events
▶ Any indicators of adverse effects of the intervention, including the following:
  – Reports of toxicity to humans/animals.
  – Environmental impacts, such as changes to the biodiversity and ecosystem due to insecticidal use.
  – Changes to the levels of phenotypic/molecular insecticide resistance, assessed using standard WHO cylinder assays and/or Centers for Disease Control and Prevention bottle bioassays/molecular techniques.
  – Changes in mosquito species composition, for example, species replacement or behaviour that reduces the efficacy of vector control interventions, for example, an increase in exophily, exophagy, zoophily or changes in biting time.

### Unintended effects
▶ Impact on human behaviour, such as changes to individuals' sleeping arrangements or use of other vector control interventions.
▶ Other vector-borne disease case incidence (eg, dengue and leishmaniasis).

### Search methods for identification of studies
We will attempt to identify all relevant trials and studies, regardless of language, date or publication status (published, unpublished, in press and in progress).

### Electronic searches
We will search the following databases: Cochrane Infectious Disease Group Specialized Register; Central Register of Controlled Trials (CENTRAL), published in the Cochrane Library; MEDLINE (OvidSP); Africa-Wide Information (EBSCO); the WHO Global Index Medicus;

the Science Citation Index Expanded (Web of Science); the Conference Proceedings Citation Index—Science (Web of Science); Embase (OvidSP); Global Health (CABI); and LILACS. We will also search the WHO International Clinical Trials Registry Platform (www.who.int/ictrp/search/en), ClinicalTrials.gov (www.clinicaltrials.gov) and the ISRCTN registry (www.isrctn.com) to identify ongoing trials.

### Searching other resources

We will contact past and present personnel at key stakeholders, donors, global policy makers, technical groups, NGOs, manufacturers, implementers and researchers in the field to identify sources of unpublished data. We will also check the reference lists of studies identified by the above methods.

### Data collection and analysis
#### Selection of studies

Study selection will follow the Preferred Reporting Items for Systematic Reviews and Meta-Analysis (PRISMA) guidelines. Two review authors will independently assess the titles and abstracts of studies identified by the literature searches. These two review authors will assess full-text copies of potentially relevant articles based on inclusion criteria. Studies will be eligible for inclusion in the systematic review and meta-analysis if they satisfy all criteria: the study population consisted of individuals of either all age groups, or children of specified age groups (ie, <5 years, >5–15 years), affected by humanitarian emergencies (any phase) in malaria endemic regions; any malaria-specific vector control intervention was evaluated; and the primary outcome of interest was malaria infection risk, measured as case incidence, infection incidence or parasite prevalence. Studies will be eligible for inclusion in secondary analyses, where none of the primary malaria risk indicators were measured, but at least one of the following secondary outcomes were reported: all-case mortality, severe malaria, anaemia prevalence, entomological inoculation rate, adult mosquito density, sporozoite rate, intervention durability, occurrence of adverse events and user acceptability/usage.

We will compare the results of our assessments and resolve any disagreements by discussion and consensus, with arbitration by a third review author, if necessary. We will ensure that multiple publications of the same study are included once. We will list excluded studies, together with their reasons for exclusion, in a 'Characteristics of excluded studies' table; any highly relevant studies which did not meet the inclusion criteria, may be included in a separate annex. The results of the study selection process will be illustrated in a PRISMA diagram.

#### Data extraction and management

Two review authors will independently extract information from the included studies into a pre-piloted database, in which study settings, methods and results from identified studies will be recorded. Disagreements in data extraction will be resolved by discussion and consensus between the two review authors and arbitration by a third review author, if necessary. In case of missing data, we will contact the original study author(s) for clarification. Data will be extracted on the following:

► Lead author and publication year.
► Study period.
► Study design: type of study; method of participant selection (for c-RCTs: adjustment for clustering); cluster size; number of clusters; cross-over; sample size.
► Study participants: population size; origins and length of time in setting (with respect to the particular emergency situation); characteristics: age, sex, ethnicity, recruitment rates, withdrawal, loss to follow-up, socioeconomic status; levels of prior malaria exposure; proportion of vulnerable individuals; prevalence of comorbidities.
► Humanitarian emergency context: geographic country or area affected; underlying cause(s) of emergency and relevant mitigating factors; structure and layout of residence area (eg, formal camp, informal settlement, open/closed); type of accommodation or housing available in study area; displacement dynamics and mobility; road access and condition; details of the political and logistical constraints encountered during different phases of humanitarian emergencies and their impact on the feasibility and choice of vector control intervention(s).
► Epidemiological context: malaria transmission intensity; incidence; prevalence; *Plasmodium* species.
► Intervention: description of intervention (active ingredient, dose, formulation, intervention format, frequency and timing of implementation); coverage, usage and adherence; any reports of refusals and reasons why.
► Cointerventions: details of any other malaria or vector-specific control interventions (such as mass drug administration, active case detection, health education and untreated bednets); coverage and adherence; any reports of refusals and reasons why.
► Delivery mechanism(s): route of intervention deployment (eg, community-wide dissemination, health facility-based distributions, integration with other essential services); any additional information related to the method of implementation, including who implemented the intervention and who was targeted (the general population or a subpopulation).
► Outcome measurements: definition of outcome; diagnostic or surveillance method; passive or active case detection; duration of follow-up; time points at which outcomes were assessed; number of events; number of participants or unit time; statistical power; unit of analysis; incomplete outcomes/missing data.
► Other: primary and secondary vector(s) species; vector(s) behaviour (population seasonality, adult habitat, peaking biting times, exophilic/endophilic, exophagic/endophagic, anthropophilic/zoophilic);

method of vector collection(s); baseline measures of phenotypic insecticide resistance and any measures during the study period; baseline measures of genotypic insecticide resistance and any measure during the study period; any measure of vector sporozoite rate; identification of vector bloodmeal(s); vector parity and age structure; any measure of user acceptability collected during the trial and reported by the treatment arm, including cross-sectional survey data of reported acceptability and qualitative data on opinions about the intervention.

► Identification of major limitations in design, analysis, results, and any aspects that would limit generalisability of results.

► Source of funding of the study; reported conflicts of interest of authors.

For dichotomous outcomes, we will extract the number of participants experiencing each outcome and number of participants in each study group. For continuous outcomes, we will extract number of participants in each study group, the mean and a measure of variance (eg, SD or SE). For count/rate data outcomes, we will extract the number of outcomes in the treatment and control groups, and the total person time at risk in each group (or the rate ratio), and a measure of variance. For c-RCTs, we will record the number of clusters randomised, number of clusters analysed, measure of effect for example, risk ratio, OR or mean difference, with CIs or SDs, number of participants and the intracluster correlation coefficient (ICC) value. For non-randomised studies, we will extract adjusted measures of effects that attempt to control for confounding. If there are no adjusted measures reported, then we will extract unadjusted measures. We will group studies for analyses according to the intervention under evaluation.

### Assessment of risk of bias in included studies

Two review authors will independently assess the risk of bias for each study, consulting a third review author if necessary. For c-RCTs, we will use the Cochrane 'Risk of bias' tool, as well as the six additional criteria listed in Section 23.1.2 of the *Cochrane handbook for systematic reviews of interventions* that relate specifically to c-RCTs.[47] For randomised cross-over trials, we will use the Cochrane 'Risk of bias' tool, as well as the additional criteria listed in Section 23.2.3 of the *Cochrane handbook for systematic reviews of interventions* that relate specifically to randomised cross-over trials. For non-randomised controlled studies, we will use the Cochrane Effective Practice and Organisation of Care 'Risk of bias' tool. Biases will be classified as 'low', 'high' or 'unclear' and summarised in graphs. For observational studies, we will use the Newcastle-Ottawa Scale .[48] We will assess the risk of bias through a hierarchy of domains, starting with 'critical', then 'serious', 'moderate' and 'low'. If any domain reaches critical risk of bias, then we will not continue with the assessment. We will make a risk of bias assessment for each outcome. If risk of bias is critical it will be excluded from the review.

Highly relevant studies with a critical risk of bias may be included in a separate annex.

### Measures of treatment effect

We will compare intervention and control data using (preferably) risk ratios or ORs if the outcome is dichotomous. For continuous data, we will present the mean difference (when studies use the same outcome measurement scale, and Standardized mean difference (SMD) when studies use different outcome measurement scale); and for count/rate data, we will use rate ratios. We will use adjusted measures of effect to summarise treatment effects from non-randomised studies. We will present all results with their associated 95% CIs. We will report any accounts that signal adverse effects.

### Unit of analysis issues

For c-RCTs, or non-randomised cluster trials, we will extract adjusted measures of effect where possible. If the study authors did not perform any adjustment for clustering, we will adjust the raw data ourselves using an ICC value. If an ICC is not reported, we will estimate this, with reference to similar studies, if possible. We will not present results from c-RCTs that are not adjusted for clustering. If we estimate the ICC, we will perform sensitivity analyses to investigate the robustness of our analyses. If we identify studies for inclusion that have multiple intervention arms, we will include data from these studies by either combining treatment arms, or by splitting the control group so that we only include these participants in the meta-analysis once. We will consider the level at which randomisation occurred, such as cross-over trials, c-RCTs, and multiple observations for the same outcome. For randomised cross-over trials, where neither carry-over nor period effects are identified as problematic, we will use a paired t-test for the analysis of continuous data from two-period, two-armed cross-over trials.

### Dealing with missing data

In case of missing data, we will apply available-case analysis and will include data on known results, as well as 'non-adherent' participants, excluded from the primary study authors' analysis, but for whom data are available.[49] 'Non-adherent' participant data may be included in 'intention-to-treat' or 'as-treated' analyses, as appropriate.[49] The denominator will vary according to the total number of participants who had data recorded for the specific outcome as well as the total number of excluded participants included per analysis type. For outcomes with no missing data, we plan to perform analyses on an 'intention-to-treat' basis. We will include all participants randomised to each group in the analyses and will analyse participants in the group to which they were randomised. If further data are needed, then the respective authors, companies or organisations will be contacted to obtain the missing data.

## Assessment of heterogeneity

Heterogeneity will be assessed by visually examining the overlap of study CIs, applying $\chi^2$ and $I^2$ statistics. A $\chi^2$ test result with a p value <0.1 will indicate significant heterogeneity. Interpretation of $I^2$ will follow the classification: 30%– 60%: may represent moderate heterogeneity; 50%–90%: substantial heterogeneity; 75%–100%: considerable heterogeneity. If we observe substantial heterogeneity we will perform further subgroup analysis. If appropriate we will explore clinical and methodological heterogeneity through consideration of the trial populations, methods, and interventions and by visualisation of trial results.

## Assessment of reporting biases

We will construct a funnel plot to assess reporting biases when performing an analysis of 10 or more studies. We will explore reasons for funnel plot asymmetry, including true heterogeneity of the effect with respect to trial size, poor methodological design (and hence bias of small trials), and publication bias, and will interpret results accordingly.

## Data synthesis

We will analyse data using Review Manager 5.[50] We will use fixed-effect meta-analysis to combine data if heterogeneity is absent. If considerable heterogeneity is present, we will combine data using random-effects meta-analysis and report an average treatment effect. We will decide whether to use fixed-effect or random-effects models based on the consideration of clinical and methodological heterogeneity between trials.

## Certainty of the evidence

We will assess the certainty of evidence using the GRADE approach.[51] Evidence will start as high-certainty. However, if there is bias due to confounding, missing data or selection of the report result, outcomes will be marked as moderate, high or unclear risk of bias and the evidence will be rated down accordingly to low-certainty evidence.

For c-RCTs, we will rate each important outcome as described by.[52] c-RCTs start as high quality evidence but can be downgraded if there are valid reasons within the following five categories: risk of bias, imprecision, inconsistency, indirectness and publication bias. Non-randomised studies can also be upgraded if there is a large effect, and if all plausible residual confounding would reduce a demonstrated effect or would suggest a spurious effect if no effect was observed.[52] We will summarise our findings in a 'Summary of findings' table.

We will summarise qualitative findings on consumer views for example, user acceptability narratively. If there are a sufficient number of included studies, two review authors will independently code the studies, and use thematic synthesis to identify themes and subthemes.

## Subgroup analysis and investigation of heterogeneity

We will stratify the analysis by study design, phase of humanitarian emergency (acute/post-acute, protracted and recovery/post-conflict phases) and vector control intervention . Where substantial heterogeneity exists, we plan to perform the following subgroup analysis:

► Intervention coverage (appropriate levels to be determined with respect to the primary mode of action of each intervention, eg, community-wide mass effect vs personal protection).
► Participants (<5 years, pregnant women, adult, mixed age groups).
► Geographical region.
► Malaria vector species behaviour.
► Malaria transmission dynamics/seasonality.

## Sensitivity analysis

We will perform sensitivity analysis on the primary outcome, to observe the effect of excluding trials at high risk of bias (for incomplete outcome data). If the ICC value is estimated, we will undertake sensitivity analyses to investigate the impact of varying the ICC value on meta-analysis results.

## PATIENT AND PUBLIC INVOLVEMENT

No patients were involved in the design of this study.

## ETHICS AND DISSEMINATION

This systematic review and meta-analysis will consist of secondary analyses of existing anonymous data and meets the criteria for waiver of ethics review by the London School of Hygiene and Tropical Medicine. As this research is based on previously published data, there will be no patient and public involvement in the design, interpretation or dissemination of study findings. This study protocol was drafted according to the PRISMA Protocols checklist.[53] Any amendments to this protocol will be documented and published alongside the results of the systematic review and meta-analysis. Study findings will be disseminated via open-access publications in peer-reviewed journals, presentations to stakeholders and relevant national and international policy makers and will contribute to the latest WHO guidelines for malaria control during humanitarian emergencies.

**Contributors** LAM, JF-A, BP and MR wrote the full text of the protocol. All review authors read and approved the final protocol draft.

**Funding** This work was supported by the WHO Global Malaria Programme (WHO-GMP). LAM, JF-A and MR are supported by the DFID/MRC/NIHR/Wellcome Trust Joint Global Health Trials Scheme. LAM and MR are also supported by UNITAID/Global Fund "New Nets Project" and FCDO-RPC "Resilience Against Future Threats through Vector Control". BP is supported by the LSHTM-Nagasaki University "Doctoral Program for World-leading Innovative and Smart Education for Global Health". We are very grateful to Jane Falconer, Information Specialist, at the London School of Hygiene and Tropical Medicine, for help with the literature search strategy, and to Drs Jan Kolaczinski, Jennifer Stevenson and Elie Akl (WHO-GMP) and Drs Natacha Protopopoff and Natasha Howard (LSHTM) for valuable protocol suggestions.

**Map disclaimer** The inclusion of any map (including the depiction of any boundaries therein), or of any geographic or locational reference, does not imply the expression of any opinion whatsoever on the part of BMJ concerning the legal

status of any country, territory, jurisdiction or area or of its authorities. Any such expression remains solely that of the relevant source and is not endorsed by BMJ. Maps are provided without any warranty of any kind, either express or implied.

**Competing interests** None declared.

**Patient and public involvement** Patients and/or the public were not involved in the design, or conduct, or reporting, or dissemination plans of this research.

**Patient consent for publication** Not required.

**Provenance and peer review** Not commissioned; externally peer reviewed.

**ORCID iD**
Louisa Alexandra Messenger http://orcid.org/0000-0002-3107-6214

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
