## [Reviewer comments · BMJ Open]

ARTICLE DETAILS

TITLE (PROVISIONAL)	Vector control for malaria prevention during humanitarian emergencies: protocol for a systematic review and meta-analysis
AUTHORS	Messenger, Louisa ; Furnival-Adams, Joanna; Pelloquin, Bethanie; Rowland, Mark

VERSION 1 – REVIEW

REVIEWER	Allan, Richard The MENTOR Initiative, Office of the Director
REVIEW RETURNED	06-Jan-2021

GENERAL COMMENTS	8. References: It is challenging to include general malaria prevention tool study references for a study in which aims to review the relative importance of malaria prevention tool studies specifically in humanitarian emergency settings. The papers selected in the reference section have been used to make strong statements in the Introduction section, as to the efficacy and suitability of IRS and or LLINs in humanitarian settings (refugee camps and more permanent housing settings), statements which in some cases are supported by a single study alone. This risks presenting and author bias, that one can make prior conclusions about two tools effectiveness in the study context (humanitarian settings) which the authors protocol is designed to study. It would be better to avoid making statements or potential judgements re: LLIN or IRS effectiveness or suitability in this protocols introduction. 4. Are the methods described sufficiently to allow the study to be repeated? Generally this protocol is very clearly developed and very competent to identify and answer the study question. However, the protocol does not include a description of the Inclusion Criteria & Exclusion Criteria by which the many papers reviewed, will be included or not. This is an important oversight, as it leaves the study authors to subsequently self determine what criteria will be used to judge if a paper merits inclusion or not. The authors are all respected academics, but are not widely field experienced in humanitarian crises settings, and if clear and balanced inclusion criteria are not preset and published, they may inadvertently risk excluding valid papers from humanitarian settings that may or may not be recognised by them as such. This might include overlooking heavy rains and localised floods/cyclones and lower levels of population displacement and epidemics in settings such as those which have occurred often in N and NE Kenya,
--

	Burma/Myanmar, Mozambique, Philippines, Ethiopia, Sudan, etc. or those settings with remote areas of the country subject to Islamic extremist groups attaching and displacing communities, such as northern Mozambique, N/NE Burkina Faso, Northern Nigeria, Niger etc. It is very important that results from published and grey literature resulting from all such humanitarian crises are included, and not unintentionally excluded.
REVIEWER	Murphy, Richard
REVIEW RETURNED	Department of Medicine, Albert Einstein College of Medicine 06-Apr-2021
GENERAL COMMENTS	This is a very well-written protocol by clear well-informed and thoughtful authors. However it is my opinion that the protocol for a meta-analysis does not merit publication for several reasons. First, a significant proportion of proposed meta-analyses are never completed. Second, the PROSPERO site exists (https://www.crd.york.ac.uk/prospero/) such that meta-analyses can be registered and the methodology elaborated. I recommend that the authors post the planned meta-analysis on that site.
REVIEWER	Sun, Chenyu
REVIEW RETURNED	AMITA Health Saint Joseph Hospital Chicago 28-Apr-2021
GENERAL COMMENTS	This protocol is well written and I am hoping to see the final results of your systemic review after it is completed. This protocol should be accepted.

VERSION 1 – AUTHOR RESPONSE

Reviewer: 1

Dr. Richard Allan, The MENTOR Initiative

Comments to the Author: [Editor - please note that the numbers and statements after the numbers relate to a checklist that we provide for reviewers to encourage them to consider all aspects of the paper - the reviewer is indicating only two sections require changes]

8. References:

It is challenging to include general malaria prevention tool study references for a study in which aims to review the relative importance of malaria prevention tool studies specifically in humanitarian emergency settings. The papers selected in the reference section have been used to make strong statements in the Introduction section, as to the efficacy and suitability of IRS and or LLINs in humanitarian settings (refugee camps and more permanent housing settings), statements which in some cases are supported by a single study alone. This risks presenting and author bias, that one can make prior conclusions about two tools effectiveness in the study context (humanitarian settings) which the authors protocol is designed to study. It would be better to avoid making statements or potential judgements re: LLIN or IRS effectiveness or suitability in this protocols introduction.

RESPONSE: our statements regarding the efficacy and suitability of IRS and LLINs for use in humanitarian settings are based on the most up to date World Health Organization recommendations (<https://apps.who.int/iris/bitstream/handle/10665/310862/9789241550499-eng.pdf?ua=1>). We have

modified the language now to try and tone this part of the introduction down. We have also added additional 15 references for studies conducted in humanitarian emergencies to try and balance this section. Added references:

1. Dolan G, ter Kuile FO, Jacoutot V, White NJ, Luxemburger C, Malankirii L, Chongsuphajaisiddhi T, Nosten F. Bed nets for the prevention of malaria and anaemia in pregnancy. *Trans R Soc Trop Med Hyg* 1993;87:620-26.
2. Luxemburger C, Perea WA, Delmas G, Pruja C, Pecoul B, Moren A. Permethrin-impregnated bed nets for the prevention of malaria in schoolchildren on the Thai-Burmease border. *Trans R Soc Trop Med Hyg* 1994;88:155-59.
3. Rowland M, Bouma M, Ducornez D, Durrani N, Rozendaal J, Schapira A, Sondorp E. Pyrethroid-impregnated bed nets for personal protection against malaria for Afghan refugees. *Trans R Soc Trop Med Hyg* 1996;90:357-61.
4. Brooks HM, Jean Paul MK, Claude KM, Mocanu V, Hawkes MT. Use and disuse of malaria bed nets in an internally displaced persons camp in the Democratic Republic of the Congo: a mixed methods study. *PLoS One* 2017; 12(9):e0185290.
5. Brooks HM, Jean Paul MK, Claude KM, Houston S, Hawkes MT. Malaria in an internally displaced persons camp in the Democratic Republic of the Congo. *Clin Infect Dis* 2017;65(3):529-30.
6. Chan CW, Iata H, Yaviong J, Kalkoa M, Yamar S, Taleo G, Isozumi R, Fukui M, Aoyama F, Pomer A, Dancause KN, Kaneko A. Surveillance for malaria outbreak on malaria-eliminating islands in Tafea Province, Vanuatu after Tropical Cyclone Pam in 2015. *Epidemiol Infect* 2017;145:41-5.
7. Protopopoff N, Van Herp M, Maes P, Reid T, Baza D, D'Alessandro U, Van Bortel W, Cossemans M. Vector control in a malaria epidemic occurring within a complex emergency situation in Burundi: a case study. *Malar J* 2007;6:93.
8. Ma C, Claude MK, Kibendelwa ZT, Brooks H, Zheng X, Hawkes M. Is maternal education a social vaccine for childhood malaria infection? A cross-sectional study from war-torn Democratic Republic of Congo. *Pathog Glob Health* 2017;111(2):98-106.
9. Charchuk R, Jean Paul MK, Claude KM, Houston S, Hawkes MT. Burden of malaria is higher among children in an internal displacement camp compared to a neighbouring village in the Democratic Republic of the Congo. *Malar J* 2016;15:431.
10. Rowland M, Webster J, Saleh P, Chandramohan D, Freeman T, Percy B, Durrani N, Rab A, Mohammed N. Prevention of malaria in Afghanistan through social marketing of insecticide-treated nets: evaluation of coverage and effectiveness by cross-sectional surveys and passive surveillance. *Trop Med Int Health* 2002;7(10):813-22.
11. Zhao Y, Zeng J, Zhao Y, Liu Q, He Y, Zhang J, Yang Z, Fan Q, Wang Q, Cui L, Cao Y. Risk factors for asymptomatic malaria infections from seasonal cross-sectional surveys along the China-Myanmar border. *Malar J* 2018;17:247.
12. Wahid S, Stresman GH, Kamal SS, Sepulveda N, Kleinschmidt I, Bousema T, Drakeley C. Heterogenous malaria transmission in long-term Afghan refugee populations: a cross-sectional study in five refugee camps in northern Pakistan. *Malar J* 2016;15:245.
13. Rowland M, Hewitt S, Durrani N. Prevalence of malaria in Afghan refugee villages in Pakistan sprayed with lambda-cyhalothrin or malathion. *Trans R Soc Trop Med Hyg* 1994;88:378-79.
14. Rowland M, Hewitt S, Durrani N, Bano N, Wirtz R. Transmission and control of vivax malaria in Afghan refugee settlements in Pakistan. *Trans R Soc Trop Med Hyg* 1997;91:252-55.
15. McGready R, Simpson JA, Htway M, White NJ, Nosten F, Lindsay SW. A double-blind randomized therapeutic trial of insect repellents for the prevention of malaria in pregnancy. *Trans R Soc Trop Med Hyg* 2001;95:137-38.

4. Are the methods described sufficiently to allow the study to be repeated?

Generally this protocol is very clearly developed and very competent to identify and answer the study question. However, the protocol does not include a description of the Inclusion Criteria & Exclusion Criteria by which the many papers reviewed, will be included or not. This is an important oversight, as it leaves the study authors to subsequently self determine what criteria will be used to judge if a paper merits inclusion or not. The authors are all respected academics, but are not widely field experienced in humanitarian crises settings, and if clear and balanced inclusion criteria are not preset and published, they may inadvertently risk excluding valid papers from humanitarian settings that may or may not be recognised by them as such. This might include overlooking heavy rains and localised floods/cyclones and lower levels of population displacement and epidemics in settings such as those which have occurred often in N and NE Kenya, Burma/Myanmar, Mozambique, Philippines, Ethiopia, Sudan, etc. or those settings with remote areas of the country subject to Islamic extremist groups attaching and displacing communities, such as northern Mozambique, N/NE Burkina Faso, Northern Nigeria, Niger etc. It is very important that results from published and grey literature resulting from all such humanitarian crises are included, and not unintentionally excluded.

RESPONSE: we completely agree with the reviewer. We have now added our explicit exclusion/inclusion criteria to lines 225-236 and 350-360, which includes an expanded definition of the humanitarian emergency context.

Reviewer: 2

Dr. Richard Murphy, Department of Medicine, Albert Einstein College of Medicine

Comments to the Author:

This is a very well-written protocol by clear well-informed and thoughtful authors.

However it is my opinion that the protocol for a meta-analysis does not merit publication for several reasons. First, a significant proportion of proposed meta-analyses are never completed. Second, the PROSPERO site exists (<https://www.crd.york.ac.uk/prospero/>) such that meta-analyses can be registered and the methodology elaborated. I recommend that the authors post the planned meta-analysis on that site. [Editor: obviously we do not follow this view as we do publish protocols].

RESPONSE: we thank the reviewer for their very kind comments regarding the quality of our manuscript. We have registered our review in PROSPERO, as it is documented in this drafted protocol, and have every intention to complete this review per protocol, since we are under contractual obligation to do so by the World Health Organization.

Reviewer: 3

Dr. Chenyu Sun, AMITA Health Saint Joseph Hospital Chicago

Comments to the Author:

This protocol is well written and I am hoping to see the final results of your systemic review after it is completed. This protocol should be accepted.

RESPONSE: we thank the reviewer for their very kind comments.